# The Evolving Role of Mucosal Histology in the Evaluation of Pediatric Functional Dyspepsia: A Review

**Craig A. Friesen [1],\*, Jennifer M. Colombo [1] and Jennifer V. Schurman [2]**

[1] Division of Gastroenterology, Hepatology, and Nutrition, Children's Mercy Kansas City, 2401 Gillham Road, Kansas City, MI 64108, USA; jmcolombo@cmh.edu

[2] Division of Developmental and Behavioral Sciences, Children's Mercy Kansas City, 2401 Gillham Road, Kansas City, MI 64108, USA; jschurman@cmh.edu

\* Correspondence: cfriesen@cmh.edu; Tel.: +1-816-302-3065; Fax: +1-816-302-9735

**Abstract:** Although not required to establish the diagnosis, endoscopy with mucosal biopsy is commonly performed in the evaluation of children with dyspepsia. Traditionally, esophagogastroduodenoscopy (EGD) has been performed in children with abdominal pain to identify pathology or conversely, to "rule-out" organic disease in order to establish a diagnosis of FD. In this review, we discuss the current diagnostic yield of endoscopically-obtained biopsies in identifying disease in children and adolescents with dyspepsia including an expanded discussion of common histologic diagnoses where clinical significance has not been definitively established. In turn, we discuss the transition of endoscopy from a search for disease to a search for biologic contributors to symptom generation, while considering the growing evidence linking non-diagnostic mucosal inflammation to FD, specifically mast cells and eosinophils.

**Keywords:** abdominal pain; functional dyspepsia; endoscopy; mast cells; eosinophils

## 1. Introduction

Chronic or recurrent abdominal pain is a critical public health issue, estimated to affect up to 19% of the pediatric population [1]. In one study, 38% of school-age children in the United States reported abdominal pain occurring at least weekly, with 24% reporting pain persisting at least 8 weeks [2]. The personal and social costs of chronic abdominal pain are high, and no clinical practice guidelines yet exist for the assessment and management of this condition [3,4].

The majority of children with chronic abdominal pain will report symptoms that fit into specific diagnoses under the broad heading of functional gastrointestinal disorders (FGIDs), as defined by Rome criteria [5,6]. FGIDs are defined by symptom complexes that occur in the absence of organic, systemic, or metabolic diseases that are likely to explain the symptoms. After initial development in adults, FGID criteria were first defined in children by a group of experts in 1999 forming the Rome II criteria [7]. These pediatric criteria were subsequently revised in 2006 (Rome III) and 2016 (Rome IV) [5,6]. There are four pain-associated FGIDs, with functional dyspepsia (FD) being one of the two most common [8]. Under Rome III, FD in children was defined by upper abdominal pain or discomfort generally not associated with a change in stool form or frequency or relieved by having a stool [5]. Under Rome IV, these criteria were significantly altered with adoption of adult criteria, resulting in FD being defined by early satiety, postprandial fullness, epigastric pain, and/or epigastric burning [6].

FGIDs were intended to be positive diagnoses based on symptom profiles that would not require a work-up to rule out other conditions in order to establish the FGID diagnosis. However, there is

ongoing debate regarding whether endoscopy should be a part of the diagnostic evaluation of FD. It has been suggested that endoscopy should be limited to patients with "red flag" symptoms to increase the diagnostic yield. However, there are no "red flags" which have been established as sufficient predictors of endoscopic findings or yield in patients with dyspepsia [9–13]. In practice, such testing is commonly performed in children with abdominal pain, including upper endoscopy with esophageal, gastric antral, and duodenal biopsies [14].

The inconsistency between criteria and practice creates some challenges and discrepancies in how FGIDs are diagnosed. Generally, mild histological changes on biopsy do not preclude an FGID diagnosis if the findings are not believed to account for the patient's symptoms. Examples would include histologic esophagitis and *Helicobacter pylori* (HP) colonization, where current pediatric guidelines recommend against considering these as entities that account for symptoms [15,16]. However, a survey of pediatric gastroenterologists indicates that three-fourths believe that histologic esophagitis and HP are indicative of organic disease that would explain a patient's pain, thus disqualifying the patient from an FGID diagnosis in their view [14]. Notably, Thakkar and colleagues have shown that approximately two-thirds of patients with abdominal pain will have a change in the care plan as a direct result of endoscopic findings, and that medical treatment for disorders identified with endoscopy are effective in two-thirds of patients [9,17]. In the end, perhaps it is less important to understand whether a specific set of histologic findings disqualify a patient from receiving an FGID diagnosis than it is to understand how endoscopic findings might help to inform the development of a comprehensive treatment plan.

Endoscopy is not without risks, and those need to be considered in the decision of whether to obtain biopsies to evaluate patients with dyspepsia. The literature supports that, overall, endoscopy is a safe procedure in children with a low rate of complications, mainly comprised of respiratory complications which are easily managed. A study of 10,236 EGDs in children reported from the Pediatric Endoscopy Database System Clinical Outcomes Research Initiative (Peds-CORI) demonstrated an overall complication rate of 2.3% [18]. Reversible hypoxia accounted for two-thirds of the complications. The risk of a cardiopulmonary complication was 5.3 times higher with intravenous sedation as opposed to general anesthesia. A study of 12,030 endoscopies (66.3% EGDs), largely performed with propofol (96.9%), demonstrated an adverse event rate of 4.8% [19]. Respiratory events occurred in 4%, and none required cardiopulmonary resuscitation. Events were more frequent in those less than 5 years of age and in those with obesity or lower respiratory disease. Lastly, a study of 217,817 children undergoing endoscopy which utilized the Pediatric Health Information System (PHIS) database found that 0.05% were admitted to the hospital and that 0.60% of patients presented to the emergency department following elective endoscopy [20]. One-half of these were for gastrointestinal symptoms. Seventeen percent were seen for respiratory symptoms and 8% for bleeding. There were no deaths. Particularly, when performed with general anesthetics such as propofol, endoscopy has a low rate of significant complications, but a patient's age, weight, and co-morbid conditions need to be considered in assessing risk. Likewise, the clinician should also consider that the procedure itself may promote anxiety and stress in the patient, which may also exacerbate abdominal pain at least in the short term.

Given that endoscopy is considered both invasive and costly, it seems important to understand the potential value of such testing, in order to determine whether the offset is sufficiently high [21]. In this review, we discuss the current diagnostic yield of endoscopically obtained biopsies in identifying disease in children and adolescents with dyspepsia. In turn, we discuss the transition of endoscopy from a search for disease to a search for biological contributors to symptom generation, while considering the growing evidence linking non-diagnostic mucosal inflammation to FD.

## 2. The Search for Disease

Traditionally, EGD has been performed in children with abdominal pain to identify pathology or conversely, to "rule-out" organic disease in order to establish a diagnosis of FD. A number of studies have evaluated the diagnostic yield of EGD in children with abdominal pain. In a systematic review

of studies reporting EGD findings in children with abdominal pain between the years of 1966 and 2005, a total of 18 articles comprised of 1871 patients met study parameters [22]. Overall, 3.6% of patients had gross endoscopic disease (primarily peptic ulcer disease and 1 patient with Crohn's disease). Histologic inflammation (esophagitis, gastritis, duodenitis) varied from 23% to 93% over 15 studies. Overall, approximately 30% were positive for HP (over 14 studies) but with a wide range of 2%–63% [22].

Since 2005, we identified 6 additional studies of EGD in the evaluation of abdominal pain where outcome findings were enumerated [9–11,17,23,24]. The prevalence of abnormal diagnostic findings ranged from 35% to 69%, with the most common diagnoses being, in descending order of frequency, reflux esophagitis, *H. pylori*, eosinophilic esophagitis, peptic ulcer disease, celiac, and Crohn's disease. Overall, these accounted for 33% of the total patient population. Only two of these studies reported frequencies for histologic non-specific gastritis (frequencies of 22.3% and 49.7%), and only one reported the frequency of non-specific histologic duodenitis (3.5%) [23]. In the largest series (*N* = 1191), 38% of the patients had an organic disease identified by EGD [10]. This 38% did not include patients with non-specific gastritis or duodenitis. If they also considered gastroduodenal eosinophil densities (antrum > 10 and/or duodenum > 20), the overall rate of an abnormal study rose to 48% [10]. If histologic esophagitis, histologic chronic gastritis, and *H. pylori* were eliminated from consideration in each of these six studies, the rate of a diagnosis would have been 15% or less.

Rather than enumerating histologic diagnoses, others have evaluated mucosal inflammation with regard to severity. Canan and colleagues evaluated a group of 161 abdominal pain patients after excluding those with predominant gastroesophageal reflux disease (GERD) symptoms and an abnormal pH study [12]. They identified organic disease in 68%, which consisted of PUD, erosive esophagitis, erosive or nodular gastritis, erosive duodenitis, or moderate-severe antral gastritis. Antral histology was normal in 29%, and revealed chronic active or mild to severe chronic gastritis in 53% and mild chronic gastritis in 18%. Puzanovova et al. evaluated histology in 124 chronic abdominal pain patients and classified histologic findings as positive, equivocal, or normal [13]. Equivocal corresponded to mild findings, including mild basal hyperplasia with rare lymphocytes or eosinophils in the esophagus, rare lymphoid aggregates in the stomach, and patchy mild villous blunting and rare inflammatory cells in the duodenum. Overall, biopsies were normal in 34.7%, equivocal in 20.2%, and positive in 45.2%. The esophagus accounted for the majority of positive findings.

Overall, approximately 35%–50% of children and adolescents with abdominal pain will demonstrate histologic inflammation in the esophagus, stomach, and/or duodenum. The pathophysiological significance for many of these inflammatory conditions is not definitively established as outcome studies are lacking. As the majority of these cases are accounted for by 3 conditions—histologic esophagitis, *Helicobacter pylori*-negative chronic gastritis, and *Helicobacter pylori*-positive gastritis—they merit further discussion. It should be noted that 50% of pediatric gastroenterologists view HP (−) gastritis as an organic disease, and 70%–75% view histologic esophagitis and HP (+) gastritis as an organic disease that would account for symptoms in patients with dyspepsia [14]. Clearly, differences in definition can significantly affect perceived yield of EGD in FD.

### 2.1. Histologic Esophagitis

Histologic esophagitis is generally viewed as having limited value in diagnosing GERD in children [15]. As previously discussed, histologic esophagitis occurs frequently in patients with abdominal pain, particularly FD, however, its relationship to functional dyspepsia is largely unstudied, and its effects on clinical outcomes is unknown. Developing an understanding will be important as there is frequent overlap of FD with GERD symptoms in both children and adults, and some consider GERD to be a part of the FD complex [25,26]. In adults, histologic esophagitis is associated with GERD symptoms, and occurs at an increased frequency in subjects with erosive esophagitis, non-erosive esophagitis, or a positive pH study compared to subjects with functional heartburn or controls [27–30]. In children with FD, histologic esophagitis is associated with increased frequencies of bitter/salty

taste, belching/burping, and regurgitation [25]. Heartburn was reported by 71% of FD children with histologic esophagitis, as compared to 32% of FD children with normal esophageal histology [25]. In adults, histologic esophagitis improves in the long term for those treated with acid reduction therapy or fundoplication [31]. While histologic esophagitis is associated with symptoms believed to be of esophageal origin in both adults and children, it remains to be determined whether therapy directed at histologic esophagitis is beneficial in relieving symptoms in children with FD.

*2.2. Histologic Chronic Gastritis*

Chronic gastritis can generally be divided into those with (HP (+)) and those without (HP (−)) associated *Helicobacter pylori* (HP). While HP (+) gastritis appears to be common in the adult general population, HP (−) gastritis is not common in the adult general population (present in <10%), and also appears to be uncommon in asymptomatic children [32–35].

The etiology of HP (−) gastritis is unknown. Proton pump inhibitors (PPIs) and non-steroidal anti-inflammatory drug use, respectively, have been implicated, but findings have varied, and a cause-and-effect has not been established [35–37]. In a pediatric study, Rosas-Blum and colleagues reported an odds ratio of 2.81 for the presence of chronic gastritis in PPI-treated patients as compared to those who had not received PPIs [37]. The association appeared to be unique to chronic inflammatory cells as there were no differences in eosinophils [37]. It also remains possible that some HP (−) cases are actually HP (+) cases where HP was not identified, possibly due to treatment with PPIs. Lastly, some HP (−) gastritis cases may be post-infectious and associated with post-infectious functional dyspepsia (PI-FD). In children, there is a high prevalence of PI-FD after acute gastroenteritis [38]. PI-FD has been associated with higher histologic scores for chronic inflammation, persistence of focal T cell aggregates, and decreased CD4+ cells [39,40].

While the clinical relevance of HP (−) gastritis is unknown, chronic inflammatory cells would have the potential to result in symptoms from cytokine production, interaction with other inflammatory cells, or effects on mechanical function. The presence of chronic gastritis could also represent a biomarker of another process, such as visceral hypersensitivity. Chronic gastritis is associated with increased gastric mast cells (to be discussed later) and enterochromaffin cells, and the density of both correlates with the density of chronic inflammatory cells [39,41]. Chronic gastritis also is associated with increased tryptase expression and increased release of histamine and 5-hydroxytryptamine [39]. While there is no straightforward association of chronic gastritis or CD3+ cell density with either electrogastrography abnormalities or gastric emptying, CD3+ density does correlate with the preprandial % tachygastria in those patients with increased eosinophil density, suggesting that they may have a role in electromechanical dysfunction [42,43]. Lastly, chronic gastritis may be associated with visceral hyperalgesia as T cells produce neurotrophic factors and neuroplastic alterations appear to be associated more with chronic, as opposed to acute, inflammation [44]. Chronic gastritis is associated with increased TRPV1 and substance P (SP) receptor densities, with variable associations with calcitonin gene-related peptide (CGRP) expression, suggesting that chronic inflammation may predispose to increased visceral sensitivity [45,46].

*2.3. Helicobacter pylori*

While HP is frequently present in asymptomatic adults, large population-based studies have demonstrated increased frequency of HP in patients with FD [47]. Eradication is associated with a small but significant effect on symptoms in adults with FD, being effective in at least a subset of patients [47,48]. Eradication has been shown to be a cost-effective treatment approach in patients with epigastric pain [47]. Positive predictors of symptomatic clinical response include nodular gastritis, duodenal ulcer, and antral neutrophilic infiltration [47,49]. Initial improvement in symptoms appears to persist long-term, following HP eradication [50].

HP has been associated with antral gastritis and duodenal ulcer in children [51]. A review of 38 pediatric studies found no association between recurrent abdominal pain (RAP) and HP [52].

However, HP may be more specifically associated with FD. Sykora and colleagues demonstrated an increased prevalence of HP in children with FD defined by Rome III, in comparison to controls and those with other pain-related FGIDs [53]. In 2 of 3 studies assessing epigastric pain, HP was associated with epigastric pain [52]. Improvement in FD symptoms has been documented in several studies, but it is not clear to what degree HP eradication is responsible for improvement [54]. Current NASPGHAN guidelines recommend against testing for HP in children with a functional abdominal pain diagnosis, but further study is needed in children specifically with FD [16].

There needs to be caution in attempting to extrapolate adult HP studies to children and adolescents, as the host response appears to be different. In adults, symptomatic improvement with eradication is associated with significant improvement in inflammation [55]. Eradication is associated with a decrease in neutrophils, eosinophils, and chronic inflammatory cells, which does not occur without eradication [56]. Since improvement in inflammation appears to be important in improving symptoms, differences in inflammatory responses between adults and children may be important in determining whether the abundant adult data are relevant to children. Children tend to have a milder degree of inflammation with less acute and more chronic inflammation [57–59]. Children tend to have more HP organisms and milder degrees of neutrophils, plasma cells, and eosinophils [59,60]. Children demonstrate a more pronounced Treg response with an anti-inflammation cytokine profile, as opposed to the Th17 pro-inflammatory response which predominates in adults [16,58,59,61]. There are also different microbiome patterns with HP in adults and children, which may be an effect of the differing host inflammatory response. Specifically, adults and children have similar microbiota when HP is not present. When present, HP tends to dominate the microbiota in adults, thus decreasing richness and diversity, while the microbiome remains significantly more diverse in children [47,62].

Non-response to eradication certainly may indicate that HP is not contributing to FD pathophysiology in a given patient. It is also possible that HP may cause dyspepsia and that HP eradication may be a necessary step, but not sufficient to completely ameliorate symptoms in some patients. This could be analogous to PI-FD, where inflammation and visceral hypersensitivity can persist after eliminating the initial inciting infection. While inflammation decreases after successful eradication, over 20% of patients have been shown to still have mild chronic inflammation 5 years after treatment [63]. HP is associated with increased gastric eosinophils and chronic gastritis, whether associated with HP or not, is associated with increased antral mast cells, both of which may persist after successful eradication [64,65]. Both cell types have been implicated in FD [66]. Eosinophil chemotaxins are increased in HP, remain high after eradication, and correlate with the density of CD45R0 and MBP-positive cells [67]. Although both cell types decreased after eradication, they did not reach normal levels, even after 24 months [67]. Chronic HP demonstrates abnormal neuroendocrine activity which improves after eradication, except for SP and CGRP [68]. The increase in SP fibers correlates with immune cells and co-localizes with mast cells in inflamed tissue [69]. CGRP and SP are inversely correlated with gastric distension thresholds for discomfort and pain in HP, suggesting that HP may predispose to visceral hypersensitivity [68,70].

The body of data regarding HP in FD suggests that some patients will benefit directly from HP eradication. Others may have persistent symptoms due to continued inflammation, which may be amenable to treatment. Still others may have persistent visceral hyperalgesia either due to persistent inflammation or as a result of resolved inflammation, the latter of which may represent yet another potential therapeutic target. Although it remains theoretical at present, it stands to reason that symptomatic treatments for FD are likely to be less effective in the context of persistent, untreated inflammation in the presence of HP, or following its eradication.

## 3. The Search for Biological Contributors

Approximately 35–50% of children and adolescents with dyspepsia who undergo endoscopy will have mucosal biopsies considered normal, and these patients will be considered to have FD. It is generally accepted that FGIDs, including FD, are best understood through the biopsychosocial model. This has implications for the evaluation and treatment of patients, including a need to

consider a wide array of factors which contribute to symptom generation and propagation (Figure 1). A considerable body of literature has been published evaluating the pathophysiology of FD, furthering our understanding of the pathophysiology within the biopsychosocial model. This literature supports viewing FD as the result of varying contributions from, and interactions between, four operating systems: psychosocial, immunological, neurological, and endocrine systems (Figure 2). The wide array of environmental, microbiological, and social factors associated with FGIDs interact or communicate with these four systems, and they likely form the pathways for symptom generation. Advances in the understanding of the pathophysiological model may portend an evolving role for endoscopy as a means to identify biological contributors to symptom generation. As described earlier, for example, this could potentially involve identification, directly or indirectly, of neurological abnormalities associated with visceral hyperalgesia. Most likely, this role would be in identifying inflammatory cells which are contributing to symptoms while not indicating a specific disease. Two such cells which have been extensively studied are eosinophils and mast cells. Findings implicating these cells include demonstration of increased density, increased activation, associations with known symptom triggers, associations with other identified pathophysiological processes, and studies indicating clinical benefit from treatments targeting these cells. This literature will be reviewed briefly below.

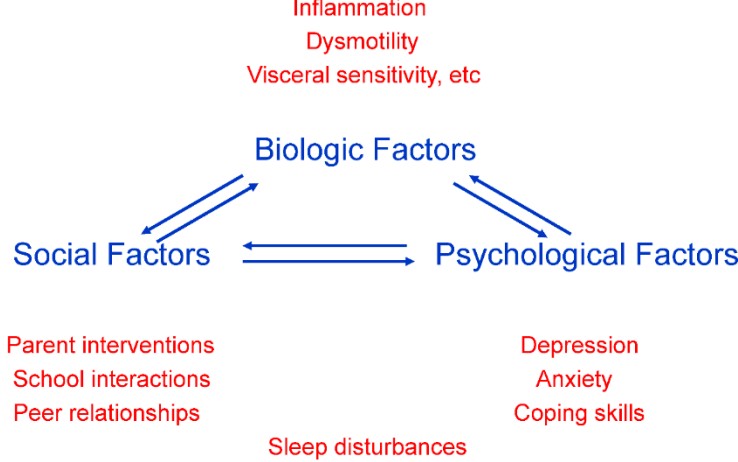

**Figure 1.** Factors contributing to pain generation in the biopsychosocial model of abdominal pain.

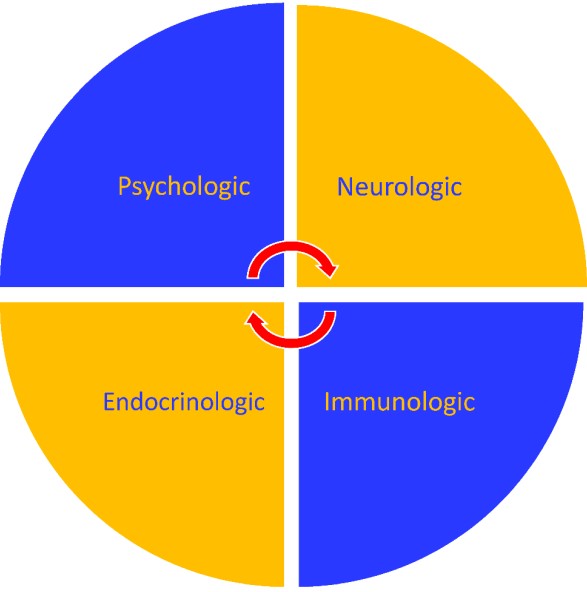

**Figure 2.** Interacting operating systems which contribute to the generation and propagation of chronic abdominal pain.

*Eosinophils and Mast Cells*

Studies continue to accumulate indicating a potential prominent role for mast cells and eosinophils in the pathophysiology of FD. At a group level, FD is associated with increased mucosal mast cell and eosinophil densities. Du and colleagues performed a systematic review and meta-analysis of microinflammation in FD, identifying 37 relevant studies published prior to May 2017 [71]. They concluded that the available evidence indicates an increase in gastric mast cells (supported by 5 studies), gastric eosinophils (9 studies), duodenal mast cells (10 studies), and duodenal eosinophils (19 studies) [71]. This association was unique to mast cells and eosinophils as mucosal intraepithelial lymphocytes, enterochromaffin cells, and neutrophils were not elevated. In a pediatric cohort, FD was associated with increased duodenal eosinophils in the absence of other histologic abnormalities [72]. As the biological effects for mast cells and eosinophils results from mediator release, potential contributions are dependent not only on cell densities but cell activation and degranulation. Increased degranulation of both eosinophils and mast cells has been demonstrated in both adults and children, further implicating these two cell types [43,73–78].

Mast cells and eosinophils are highly interactive with each other (as well as T cells) and both also exhibit self-sustaining autocrine activity. Incubation of mast cells with eosinophil products (such as major basic protein and eosinophil cationic protein) results in degranulation and histamine release [79]. Mast cells and eosinophils both produce interleukin-5 which augments mast cell cytokine production and stimulates eosinophil growth, chemotaxis, and activation [80,81]. Both also express leukotriene and TNF-$\alpha$ receptors which promote chemotaxis, survival, and activation of both cell types [80,82]. Mast cell-produced eotaxin and histamine serve as chemoattractants for eosinophils [80,82]. Given the effects of the multiple mediators produced by mast cells and eosinophils, it is likely that activation of either cell will alter the function of the other.

Mast cells and eosinophils have a number of associations with other factors which have been implicated in FD, including anxiety/stress and mechano-sensory dysfunction (Figure 3). Anxiety and stress are highly implicated in the development and/or maintenance of FGIDs, including FD. Approximately half of children have elevated anxiety scores alone or in association with more global psychosocial dysfunction [83]. Corticotropin releasing hormone (CRH) is the major mediator of the stress response. CRH receptors are expressed on mast cells (and may be expressed on eosinophils under stress conditions) [84,85]. Once CRH activates mast cells, they release mediators which recruit and activate eosinophils and both cell types interact in a bi-directional fashion with T helper cells. Anxiety scores are positively correlated with antral mast cell density in children with functional dyspepsia [86]. Under physical stress conditions, adults have demonstrated selective luminal release of tryptase and histamine at physiologically significant levels [87]. Disodium cromoglycate, a mast cell stabilizer, has been shown to reverse visceral hypersensitivity in stress-sensitive rats [88].

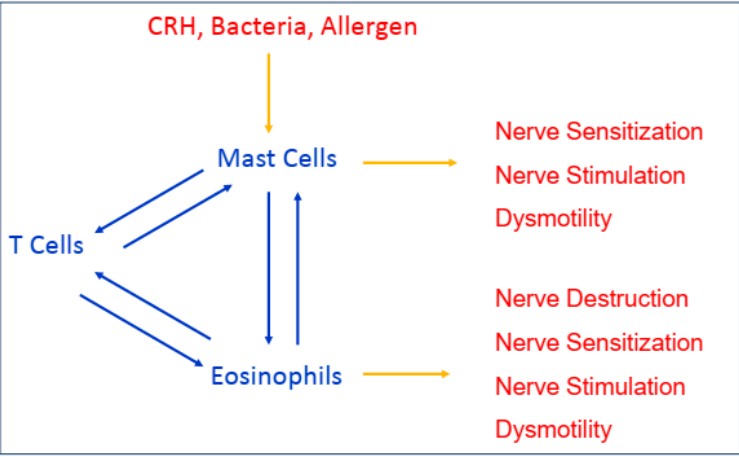

**Figure 3.** Interactions between pain triggers and inflammatory cells with resultant physiologic alterations.

Activated mast cells release cytokines which can stimulate afferent sensory nerves, affect mechanical functioning, and promote visceral hypersensitivity. Abnormal nerve function has been observed in FD in association with increased duodenal mast cells and eosinophils [89]. Histamine may be particularly important as it can stimulate afferent nerves directly, and it has been shown to sensitize TRPV1 which mediates visceral hypersensitivity [90,91]. Cysteinyl leukotrienes (cysLT) may also play an important role in increasing nerve sensitivity as they are expressed on spinal nerve terminals and have been shown to affect the contractility of the esophagus, stomach, small and large intestines, and gall bladder [92–99]. Mast cell degranulation in the proximal stomach has been demonstrated following balloon distension in patients with hypersensitivity [100]. In children, increased antral mast cell density is associated with slower gastric emptying and preprandial tachygastria [43].

There are a limited number of studies of treatments directed at mast cells or eosinophils in FD, but those that are present would suggest clinical benefit. There have been no studies of mast cell stabilizers in adults with FD. Disodium cromoglycate has been shown to modulate mast cell activity and improve symptoms in adults with irritable bowel syndrome (IBS) [101]. Likewise, ketotifen, a mast cell stabilizer which also antagonizes H1 receptors, has been shown to decrease visceral hypersensitivity in adults with IBS [102]. In an open-label study of disodium cromoglycate in children with FD and mucosal eosinophilia, resolution of pain was seen in 89% of children who failed to respond to H1 and H2 antagonism [103]. In a double-blind, placebo-controlled, crossover trial of pediatric FD associated with mucosal eosinophilia, montelukast, a cysLT antagonist, demonstrated efficacy in the relief of pain [104]. In a subgroup analysis of those patients with eosinophil densities from 20 to 29 eosinophils/high power field, the response rate was 84% with montelukast as compared to 42% with placebo [104]. The high response rate was confirmed in another study which suggested that the short-term benefit was not related to an anti-inflammatory effect, suggesting an effect on enteric nerves or motility [105]. Consistent with the suspected association between stress and inflammation, another study of pediatric dyspepsia with duodenal eosinophilia demonstrated additive clinical benefit from adding biofeedback-assisted relaxation training to the anti-eosinophil regimen [106].

At a group level, sufficient evidence exists to indicate a significant pathophysiological role for mast cells and eosinophils in FD. The more relevant question to the current discussion is what percentage of individual patients with FD and "normal" biopsies have elevated mast cell and/or eosinophil density? Normal ranges for mast cells and eosinophils are not fully established as ethical considerations preclude tissue sampling in otherwise healthy children. Most pediatric studies have defined controls by normal gross endoscopy, with normal biopsies excluding evaluation of eosinophils and mast cells. This assumes that eosinophils and mast cells cannot be the sole abnormality, though the FD literature suggests that elevations in eosinophils and mast cells are not associated with elevations of other inflammatory cells [72,75].

There is not yet sufficient data to estimate meaningful cut-offs for mast cells. In one study by Singh and colleagues, peak antral mast cell density was >15/hpf in 71%, and peak mast cell density was >20/hpf in 68%, but this must be viewed cautiously until more data are available [107]. However, existing pediatric data provide suggestive ranges for what could conservatively be considered abnormal for eosinophils. In an autopsy study (where the presence of gastrointestinal symptoms was unknown), eosinophil density was <10/hpf in 100% of antral specimens and <20/hpf in 82% of duodenal specimens [108]. A review of biopsies from 682 children undergoing diagnostic endoscopies revealed eosinophil density <10/hpf in the antrum in 90%, and eosinophil density <20/hpf in the duodenum in 93% [109]. Maximum eosinophil density in the antrum was 8/hpf, and in the duodenum was 26/hpf. Debrosse and colleagues evaluated previously obtained endoscopic specimens, and the mean + 2 standard deviations for eosinophil density was 4.5/hpf in the antrum and 20.6/hpf in the duodenum [110]. Given these findings in children being evaluated for gastrointestinal symptoms and/or suspected gastrointestinal disease, it seems reasonable (and likely conservative) to consider upper limits of normal of 10 eosinophils/hpf in the antrum and 20/hpf in the duodenum. When evaluating pediatric FD patients at the individual level using these criteria, peak antral density

was >10/hpf in 62% and peak duodenal eosinophils were >20/hpf in 67% [107]. Thus, an increase in mucosal eosinophils appears common in pediatric FD patients, and early studies suggest this may be an encouraging biological pathway to consider in treatment.

## 4. Conclusions and Future Directions

The evolution from utilizing EGD to identify not only disease but as a tool to identify biological contributors will continue to enhance the value it provides for patient care. The yield of EGD in the evaluation of abdominal pain in children appears to be high, especially if we consider that two-thirds of patients with "normal" biopsies will have elevated mucosal eosinophil (and possibly mast cell) density and that many of the identified abnormalities are amenable to treatment. There is sufficient evidence for clinicians to request that pathologists enumerate eosinophils on mucosal biopsies in patients with dyspepsia which can be done on routinely stained specimens. It remains to be determined whether enumerating mast cells, which requires special stains, will have clinical utility.

Clearly, further work is necessary to fully understand the implication of common forms of histologic inflammation (esophagitis, HP (−) chronic gastritis, and HP (+) chronic gastritis). Most importantly, treatment trials are necessary to understand the abilities of these entities to predict clinical response to medications directed at HP, inflammation, or visceral hyperalgesia. Similarly, while current evidence highly implicates mucosal mast cells and eosinophils as biological contributors in FD, and as mediators between psychological and physiological factors, more controlled treatment trials are needed in patients with FD to define the clinical benefit of various medications, particularly mast cell stabilizers.

Future work is needed to determine if symptoms (or symptom clusters) or biomarkers can be identified which are predictive of histologic findings, in an effort to decrease the need for endoscopy and the associated costs. In the end, while EGD remains costly, the costs of not adequately treating abdominal pain are also significant. These include not only the direct costs, such as emergency department visits, hospitalizations, tests other than endoscopy, and sequential medications which are often not effective, but the psychosocial costs of missed school, missed activities, and promotion of mood disorders. There are also the hidden costs to the patient's family, including family stress and conflict, as well as missed days from work. These secondary costs only increase with time, making a "wait and see" approach less attractive in a value-based, patient-centered system. The ability of EGD to frequently identify abnormalities which can be considered in the treatment program, in addition to psychosocial interventions, can be expected to significantly shift the cost balance in the favor of performing endoscopy.

**Author Contributions:** All authors contributed to the conceptualization, writing, and editing of the manuscript.

**Funding:** This research received no external funding.

**Conflicts of Interest:** The authors declare no conflict of interest with the exception that C.A.F. has received research funding from Merck & Company.

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
