# Peer review of "The Evolving Role of Mucosal Histology in the Evaluation of Pediatric Functional Dyspepsia: A Review"

_gastrointestdisord, doi:10.3390/gidisord1010013_

Reviewer 1 Report

Very interesting and well presented.

Reviewer 2 Report

The authors answered to the comments and questions of the reviewers properly.

Reviewer 3 Report

Authors have made corrections as requested by the reviewers and article looks more polished now. Minor grammatical errors need to be rechecked.

Reviewer 4 Report

The Authors have addressed all comments 

I do have no other suggestions 

This manuscript is a resubmission of an earlier submission. The following is a list of the peer review reports and author responses from that submission.

Round  1

Reviewer 1 Report

The authors have chosen a very up to date topic and made an attempt to discuss the role of endoscopy in evaluation of gastrointestinal symptoms in children. Their point of view as looking at transition of endoscopy from a search for disease to a search for biologic contributors and symptom generation is interesting and attractive. However the paper is not easy to follow. The chapters are long (e.g. 2.3 Helicobacter pylori and  3.1 Eosinophils and Mast Cells) and could be shortened. Also there is relatively scarce discussion concerning the role of GI endoscopy or novel endoscopic techniques in evaluation of symptoms. Instead there is a long read concerning the clinical and histopathologic interplays, which does not help to draw conclusions for the reading clinician. As the treatment options are still limited and the risk of serious organic disease (e.g. maligancy) in pediatric population low, the text does not convince the reader that endoscopy is a must to perform examination. I would suggest trying to shorten the discussion (e.g. regarding histopathologic findings) and adding more on the role of endoscopy itself. For example looking at and discussing studies like below:

Kiesslich R et al. Local barrier dysfunction identified by confocal laser endomicroscopy predicts relapse in inflammatory bowel disease. Gut 2012;61:1146e1153 or Fritscher-Ravens A i wsp Confocal Endomicroscopy Shows Food-Associated Changes in the Intestinal Mucosa of Patients With Irritable Bowel Syndrome.  Gastroenterology 2014; 147:1012-1020.

Also please consider adding brief discussion on the role of villous atrophy in symptom generation. According to Kyoto concensus celiac disease is not the only disease associated with villous atrophy and other factors might contribute to these findings.
Is endoscopy helpful in evaluation of patients with this finding ?

Please consider following citation and brief discussion: 

Vanheel H, Vicario M, Boesmans W, Vanuytsel T, Salvo-Romero E, Tack J, Farré R. Activation of Eosinophils and Mast Cells in Functional Dyspepsia: an Ultrastructural Evaluation. Sci Rep. 2018 Mar 29;8(1):5383

Vanheel H, Carbone F, Valvekens L, Simren M, Tornblom H, Vanuytsel T, Van Oudenhove L, Tack J. Pathophysiological Abnormalities in Functional Dyspepsia Subgroups According to the Rome III Criteria. Am J Gastroenterol. 2017 Jan;112(1):132-140

Also a short paragraph could be added regarding safety of endoscopic evaluation in children and risk of sedation. Otherwise please consider the alternative title of the paper; for example The evolving role of endoscopy and histopathology in the understanding of pediatric functional dyspepsia. 

Please check citations in the text (the year is frequently missing)

Minor spell checks

I can not find the figures attached (Figure 1 and 2 and missing)

Author Response

Thanks you for your review of the manuscript. I made the following changes to address your recommendations:

It is a point well taken that the manuscript is more related to histopathology (and particularly mucosal inflammation) than it is a review of endoscopy. Therefore we did change the title to reflect this emphasis (in addition to also changing the abstract and making some minor wording changes throughout.) as recommended. Consequently we did not expand the discussion of endoscopy and endoscopic techniques.

We included the Vanheel manuscript related to inflammation as recommended (line 279).

We added a paragraph discussing EGD complications, particularly those related to sedation (Lines 58-77).

All references have been corrected. I mistakenly submitted the manuscript version before I converted the author names to numbers as per the journal style.

Figures 1-3 are being resubmitted.

Reviewer 2 Report

General: The authors have identified an interesting research question. “The Evolving Role of Endoscopy in the Evaluation of Pediatric Functional Dyspepsia” is an interesting topic. Interesting to read. Excellently written.

All in all, the analysis in this article is performed accurately and rigorously. Grammar needs to be rechecked throughout the manuscript.

Line 338 change amendable to amenable

Although authors mentions role of endoscopy in evaluation of pediatric functional dyspepsia in the The Search for Biologic Contributors sections authors veer away from the topic discussing more about pathology than the acutal role of endoscopy

Studies on the diagnostic yield of EGD in children with chronic abdominal pain and functional dyspepsia should be included and mentioned

Risks of endoscopy should also be mentioned

Overall a nicely written review article.

Author Response

Thanks you for your review of the manuscript. I made the following changes to address your recommendations:

"amendable" was changed to "amenable" on line 365.

It is a point well taken that the manuscript is considerably more about mucosal histology itself and this was the intent. The title, abstract, and some minor word changes were made to more clearly indicate that the topic is histopathology.

We added a paragraph on the risks, particularly those related to sedation (lines 58-77).

Reviewer 3 Report

In this review, the authors attempt to examine the role and advantages of endoscopy that has been done to evaluate chronic abdominal pain in the pediatric population. This chronic pain comes under the broad heading of functional gastrointestinal disorders (FGIDs) and is typically Not associated with any organic, systemic, or metabolic diseases which may explain this symptom. Authors state that initially FGID was supposed to be diagnosed without any work up, and endoscopy was to be used for only cases associated with the red flag. However, endoscopy is used more commonly. Authors then discuss the diagnostic yield of esophagogastroduodenoscopy (EGD) at present in identifying the disease in children and adolescents with dyspepsia, as well as discuss the possible transition of endoscopy from a search for the disease to a search for potential biologic contributors to symptom generation. Authors report two such potential contributors are Two 235 such cells which have been extensively studied are eosinophils and mast cells.

Authors conclude by saying that chronic abdominal pain may bring many social and financial stressors.  The EGD may be costly but the ability of EGD to frequently identify abnormalities which can be considered in the treatment program in addition to psychosocial interventions may shift the cost balance in the favor of performing endoscopy.

Comments:

It is a good review article, addressing an important question. However, as per the journal’s criteria, it does not fulfill the Prisma checklist.

1. A known cause of abdominal pain in children is the presence of psychiatric issues such as depression or anxiety. But authors have mentioned this aspect very briefly. Considering a biopsychosocial approach to evaluate the patient is important too.  Since the inflammation present in the stomach has been associated with psychological issues.  As the following two studies suggest:

i; Hyams, Jeffrey S., et al. "Dyspepsia in children and adolescents: a prospective study." Journal of pediatric gastroenterology and nutrition 30.4 (2000): 413-418.

ii; Schurman, Jennifer V., et al. "Symptoms and subtypes in pediatric functional dyspepsia: relation to mucosal inflammation and psychological functioning." Journal of pediatric gastroenterology and nutrition 51.3 (2010): 298-303.

2.  As the authors mentioned earlier in the manuscript, cost and being an invasive procedure are two problems associated with endoscopy. However, in the conclusion section, authors only discussed cost but there is no mention of the fact that endoscopy is an invasive procedure and may itself be a cause of anxiety.

3. The abstract should be written as well.

4. It is not clear that in keywords, what ‘ma’ stands for?

Author Response

Thanks you for your review of the manuscript. I made the following changes to address your recommendations:

We expanded the discussion of the biopsychosocial model including the manuscript by Schurman, et al (lines 298-299). We could not find info regarding inflammation and psychologic function in the manuscript by Hyam, et al- perhaps it is a different manuscript. Figures 1 and 2 which did not appear were included to more full illustrate the biopsychosocial model. Figure 3  also includes the inflammatory pathway and down stream effects of immune cell activation for pain triggers including stress. 

We added a paragraph on procedure risks (lines 58-75) and added a comment regarding that EGD may be anxiety provoking (line 75-77)- a point well taken.

An abstract was included with the submission but did not appear in the draft. The keywords have been corrected.

Reviewer 4 Report

Grammatical, Typographycal errors:

Page 2, para 2, line 57: “…to understand to [potential value…”; should perhaps be: to understand the potential value…

Page 4, para 1 , line 151:  “…in those patient with increased…”; should perhaps be: in those patients with…

Page 5, para 3, line 227: …” four operating systems: psychosocial, inflammatory, neurologic and endocrine systems.”; would suggest to consider changing the ‘inflammatory’ one for immunologic, which is really above the inflammatory response and entails more of a system.

Comments:

It would be important to clarify and / or emphasize perhaps that the terms used as ‘esophagitis’, ‘gastritis’ and ‘duodenitis’, correspond to histological description of biopsy material and not to endoscopic reports, as very frequently the language used by many pediatric endoscopists, use words that normally describe inflammatory changes when all they saw was perhaps erythema.

The authors may want to mention, or comment, whether it is known or explore the possible correlation of the histological changes described and particularly the presence of higher counts of eosinophils and also activated (or not) mast-cells in the antrum and duodenum of these patients, in association with the use of medications such as PPIs or H2-blockers to decrease or control acid secretion, as a good number of not a majority of patients when they get to the gastroenterologist, have already been prescribed and are taking these medications, which definitely alter significantly the biology of those anatomic sites. Are there any studies or mention of differences in the biopsy specimens among these patients?

Would there be any particular stains we should all be asking the pathologist to perform on the biopsies provided and perhaps still insist by mentioning the importance of always obtain biopsy specimens in these patients with normal endoscopic appearance as well as the correct handling of the biopsy specimens when possible, in terms or orientation before fixation, to preserve and enhance the quality of this important source of information.

It would not be out of context to also perhaps consider mentioning briefly the importance of the use of most objective description of the endoscopic findings in a language that does not imply or comprise a histological diagnosis.

As one of the goals would be to try and develop uniform clinical as well as endoscopical parameters to determine who needs endoscopy, with the aim at trying to improve the yield of positive and ideally more specific and relevant findings on histology and be able to correlate those with other clinical testing and tools, it would be important to develop standard questionnaires as well as physical and endoscopical findings that can be given more objective value in data collection.

Author Response

Thanks you for your review of the manuscript. I made the following changes to address your recommendations:

We changed the title and abstract and made some minor wording changes throughout to indicate that histology, rather than endoscopy, is the focus of the manuscript. We made sure that "histologic" appeared before esophagitis and added "histologic" before "chronic gastritis" (e.g. section title on line 149) to indicate that we use these terms to refer to histology rather than gross appearance.

We indicated that PPI use is not associated with an increase in eosinophils (lines 158-159). The only other potentially related references we could find indicated that PPI alters mediators in allergic rhinitis and EoE in ways that would be expected to decrease eosinophils. These might be too removed from the subject but we would be happy to add them if you feel that it would enhance the manuscript. Overall, we could not find any studies indicating that acid suppression would increase eosinophils or mast cells, only manuscripts which would indicate that PPIs might decrease eosinophils.

We added a comment about pathologists counting eosinophils which can be done on routine H&E stains and that evidence does not yet support performing special stains for mast cells (lines 365-368).

We added a comment regarding the need for future work to find clinical indicators or biomarkers that would predict histologic findings (lines 377-379)